# Hub-and-Spoke Logistics Network Considering Pricing and Co-Opetition

**Jian Zhou** [1], **Kexin Xu** [1], **Yuxiu Zhao** [1], **Haoran Zheng** [2,*] **and Zhengnan Dong** [3]

1    School of Management, Shanghai University, Shanghai 200444, China; zhou_jian@shu.edu.cn (J.Z.);
     xukexin0430@shu.edu.cn (K.X.); zhaoyvxiu19960614@163.com (Y.Z.)
2    School of Economics, Shanghai University, Shanghai 200444, China
3    School of Science, Shanghai University, Shanghai 200444, China; zhengnandong@foxmail.com
*    Correspondence: haoranzheng@shu.edu.cn; Tel.: +86-21-66134414 (ext. 805)

**Abstract:** With the rapid development of the logistics market, the construction of an efficient "channel + hub + network" logistics system, that is, a hub-and-spoke logistics network, is of great importance to enterprises. In particular, how to reduce costs and increase efficiency in both the construction of network facilities and actual operations, and to formulate reasonable prices for the logistics service needs in the entire market are crucial strategies and decisions for enterprises. Under such a background, this article starts from the perspective of duopoly logistics enterprises that jointly build networks and allow the transfer of surplus capacity and carbon credits, and studies the hub-and-spoke logistics network design that also considers the relationship between service pricing and co-opetition. Considering the corporate profit and difficulty of implementation as a whole, the co-opetition is a better choice than the complete competition and perfect cooperation. In addition, the remaining capacity of the company, the transfer of carbon credits, the joint construction and sharing of hubs, and strategic cooperation in the area of corporate common pricing under the price framework agreement are conducive to the realization of an increase in corporate operating profits, a better market share and more favorable pricing.

**Keywords:** hub-and-spoke logistics network; co-opetition; service pricing; bi-objective mixed integer non-linear programming

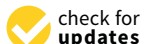



## 1. Introduction

As the "artery" of national economic development, modern logistics industry has been vigorously encouraged to develop in China these years [1]. The demand for the logistics market is relatively strong, and the potential market urgently needs further development. In view of the fact that the total logistics cost of the whole society accounts for a significantly high proportion of GDP, a convenient and efficient modern logistics service system with reasonable layout is vital to the development of the times. For a single logistics enterprise, the relatively high fixed asset investment in the initial stage of constructing a hub-and-spoke logistics network leads to a long capital recovery cycle and a small profit margin for the enterprise. It even achieves break-even after many years of operation. In addition, the situation of changing prices in the market price war and the lack of reasonable service pricing are also major breakthrough problems faced by logistics enterprises. Therefore, it is vital for the development of the logistics industry to consider how they should set prices, choose competition and cooperation models, and promote enterprises to maximize their economic profits on the premise of meeting customer service needs.

Starting from the "channel + hub + network" logistics system design of duopoly enterprises in different competition and cooperation scenarios, this paper combines the pricing of logistics services with the hub-and-spoke logistics network design to establish a mathematical programming model. The standard dataset is used to solve the network design model, and the business goals and results of the three different situations of co-opetition,

complete competition, and perfect cooperation are compared, which provide a cooperative wisdom scheme for the actual operation of logistics enterprises. The contribution of this research lies in:

- Design the mathematical programming model of hub-and-spoke logistics network considering the co-opetition. This article starts from the perspective of co-building networks, allowing enterprises to transfer surplus capacity and carbon credits, and establishes a theoretical model which is more conducive to the use of social resources and the realization of enterprise objectives;
- Combine service pricing with hub-and-spoke logistics network design, while considering price demand elasticity and the cross-sensitivity coefficient (substitution) of demand to price. The combination of market pricing, price demand relationship, and hub-and-spoke logistics network design makes the price setting more suitable for the laws of market operation;
- Compare and analyze the "channel + hub + network" logistics system design of duopoly enterprises in the three scenarios of co-opetition, complete competition, and perfect cooperation. This article considers the different competition situations of duopoly enterprises in the same market for service pricing and network design, discusses the impact of different situations on business operations, and provides decision-making support for the operation of logistics enterprises.

The rest of this article is organized as follows. Section 2 summarizes the research related to hub-and-spoke logistics network design and logistics service pricing under different competition and cooperation environments. Section 3 discusses the construction of the mathematical programming model over the design problem of the axial amplitude network of duopoly enterprises in the context of the coexistence of service pricing and senarios of cooperation, co-opetition and complete competition. Section 4 uses standard datasets to solve the model, compares experimental results and conducts sensitivity analysis. Section 5 summarizes and discusses the full text, and prospects for future research.

## 2. Literature Review

In the hub-and-spoke logistics network, the main role of the pivot point is as a transfer point in the journey from the start point to the end point, and reach the scale economy effect in logistics transportation by concentrating and dispatching goods. Regarding the scale economy effect in logistics transportation, it can be traced back to O'Kelly and Bryan [2] who applied the scale economy effect to the study of the mainline transportation system without capacity limitation. Based on this research, Ebery et al. [3] proposed the idea and solution algorithm for the multi-point allocation problem with capacity constraints. Later, this research was extended to large-scale transportation allocation problem through two to three hubs [4]. Since then, more scholars have positioned their research perspectives on how a single enterprise can achieve optimization through pivot point selection and non-pivot point allocation, such as minimizing the total operating cost of the enterprise [5,6], the completion time of the last service in the system [7,8], the transportation time and transportation cost [9], the energy consumption of the system [10], maximizing the level of corporate profit [11], maximizing traffic, and minimizing congestion at the same time [12].

The research of single-enterprise hub-and-spoke logistics network design mainly starts from the perspective of the enterprise's own operation, and considers how to better improve customer service level and business objectives through network layout [13]. However, under the market economy system, the competition between enterprises with the same service scope or providing similar products or services is fierce in order to grab more market share [14,15].

The current research on the co-opetition relationship in logistics industry mostly refers to that two or more enterprises establish a strong logistics relationship network through cooperation methods, such as information exchange and infrastructure sharing, at the same time, enterprises compete with each other to maximize their own market share. Rohaninejad et al. [16] studied the competitive relationship among potential investors

in order to obtain more suitable locations and customers in the market. Niu et al. [17] formulated competing e-commerce firms' incentives regarding logistics cooperation via logistics sharing alliance, and studied whether enterprises with disadvantages in logistics services should join logistics sharing alliance. Gao et al. [18] designed an automated negotiation model to describe both the collaborative game process among the team members and the competitive negotiation process between the allied team and the stakeholder. Nasr et al. [19] introduced factors affecting a firm's optimum supply chain innovation strategy into the study, and took a novel approach to address the dilemma of innovation sharing versus protection among supply chain partners. Song et al. [20] studied the price competition and cooperation between two hub ports from the perspective of game theory. In addition, Rezapour et al. [21] studied the competition between the new supply chain and the existing supply chain. The research involved product pricing and distribution center and retailer location issues, and carried out small-scale and large-scale solutions at the same time. Regarding the hub-and-spoke logistics network design in the context of co-opetition, Monemi et al. [22] conducted related research, but their research background is that two competing enterprises are subsidiaries of a parent enterprise. In the previous research, most of the studies only considered cooperation cost and competition cost, in addition that co-opetion relationships are common nowadays, it is necessary to study the hub and spoke logistics network design in the context of co-opetition.

The classic single-enterprise pivot point selection research does not consider the issue of service pricing when the goal is to maximize the profit level. However, in reality, the economic profit sources of enterprises mainly include two parts: service charges and cost savings. In terms of the research on service pricing and pivot point selection of logistics networks, Lüer-Villagra and Marinov [23] innovatively assume the service pricing strategy of the original enterprise in the market is the sum of cost and a fixed income ratio, and the price index form is used as the logical model of customer allocation. Kress and Pesch [24] studied the hub median problem in a competitive environment, combined service pricing, and adopted a logical model of customer allocation based on distance and price as the allocation principle. Liu et al. [25] developed a quality-based price competition model for the waste electrical and electronic equipment recycling market in a dual channel environment. Lin and Lee [26] innovatively used the inverse demand function in their research, that is, the price is expressed as a function of demand to solve the elastic demand. In addition, interval pricing, as one of the important pricing strategies, has also been applied to the problem of hub-and-spoke logistics networks. Setak et al. [27,28] adopted an interval pricing strategy in the construction of the service pricing model, and at the same time considered the selection of pivot points and the problem of vehicle path planning in different regions.

For the problem of considering both service pricing and pivot point selection in a competitive environment, scholars mainly divided the research on customer demand distribution into the following aspects: path cost [29–32], price [23,33–35], cost and price [14,16], route cost and transportation time [36–40], cost and service [41,42]. Through the review, it is found that few literature consider both cost and price factors, and the previous studies mostly take distance, cost, and time as the basic parameters. Therefore, it is inspiring for this study to take the consideration of service pricing and customer demand distribution in the design of hub-and-spoke logistics network.

By investigating the literature from the aspects of hub-and-spoke logistics network design, logistics service pricing and co-opetition, we found that hub-and-spoke networks are widely used in many aspects of life, such as telecommunications, postal distributions, emergency services, computer networks, transportation systems, etc. For hub-and-spoke logistics network, the current theoretical research mainly focuses on two aspects: single enterprise and duopoly or multiple enterprises in competitive situation. The research on the design of hub-and-spoke logistics network under the co-opetition relationship has also been paid attention to these years, but the number of studies in this area is relatively small. In addition, in practice, multi-enterprise operation is a normal market condition, and it is

necessary to study the operation decisions of multi-enterprise under different operation situations. In this case, this paper takes duopoly enterprises as the research object, and discusses the hub-and-spoke logistics network design under three scenarios of complete competition, perfect cooperation, and co-opetition. Therefore, the proposed model is more conducive to the utilization of social resources and conforms to the business objectives. What is more, the competition of market price demand, cooperation of building networks and the transfer of surplus capacity between enterprises are also considered in this paper.

## 3. Model Construction

This section first describes the research problem, and then proposes the relevant assumptions through necessary simplification of the problem. Finally the hub-and-spoke logistics network design optimization models in three scenarios are constructed and related explanations are given.

### 3.1. Problem Description and Model Assumptions

First of all, the research object of this paper is the duopoly logistics enterprises in the market. In terms of competition, it is reflected in the fact that duopoly enterprises attract consumers' demand by making logistics service prices, and then form market competition. The cooperation mainly refers to two aspects, on the one hand, it refers to the duopoly enterprises mentioned above to build logistics network of cooperation, in order to reduce the investment in fixed assets. On the other hand, it refers to allowing enterprises to transfer surplus capacity between the duopoly enterprises. Therefore, duopoly enterprises form the mode of co-opetition, and provide logistics services in the market. At the same time, according to the purpose of enterprise operation, this paper takes the profit maximization of both enterprises as the goal of model construction.

In order to facilitate the establishment of the model, three assumptions are made as follows.

Firstly, for the structure of enterprises, it is assumed that the location of demand nodes in the market is known and only duopoly enterprises enter the market. At the same time, the total service capacity of two enterprises can fully meet the potential market demand, but their abilities are different, and their services are substitutable with each other, so there is competition to obtain demand through price.

Secondly, as the role of price competition mechanism is reflected in price elasticity of demand and cross-sensitivity coefficient of demand to price, the market demand for logistics services is influenced by the prices of the duopoly enterprises. Moreover, the duopoly enterprises A and B are rational decision makers, both of which will react to each other's price adjustments.

Finally, for the "channel + hub + network"logistics system, as illustrated in Figure 1, it is assumed that non-hub points cannot transport directly with each other, while hub points can transport directly with each other; and the logistics movement is bidirectional.

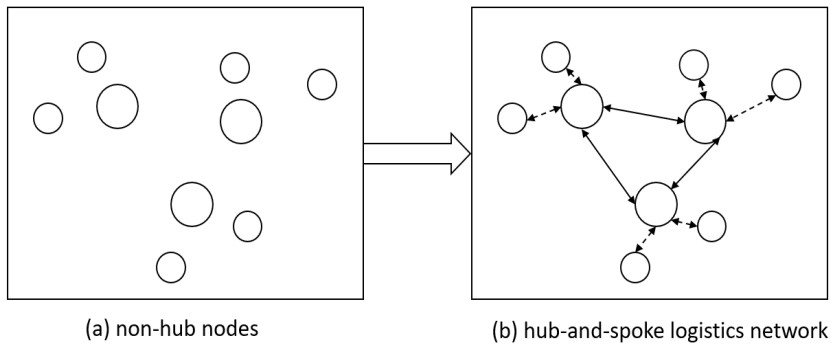

(a) non-hub nodes                    (b) hub-and-spoke logistics network

**Figure 1.** Hub-and-spoke logistics network diagram.

Based on the research problem and assumptions, the set, parameters and variables are defined as Table 1.

**Table 1.** Sets, parameters, and variables in model.

| **Decision Variables** | |
|---|---|
| $P_{ij}^{A(B)}$ | the unit price charged by enterprise A(B) for providing one unit of logistics service |
| $D_{ij}^{A(B)}$ | the amount of service demand from node $i$ to node $j$ satisfied by enterprise A(B) |
| $ta_{ij}$ | the total transfer of excess capacity between enterprises from node $i$ to node $j$ |
| $E$ | the total transfer of excess carbon credits among enterprises |
| $Y_k^{A(B)}$ | the 0-1 variable that measures whether the hub point $k$ is built by enterprise A or B. If it is built by enterprise A(B), the corresponding value is 1; otherwise 0 |
| $y_{ik}^{A(B)}$ | the 0-1 variable that measures whether the non-hub point $i$ is assigned to hub point $k$ for enterprise A or B. If it is assigned for enterprise A(B), the corresponding value is 1; otherwise 0 |
| **Sets** | |
| $G$ | the set of whole network, $G = (N, R)$ |
| $N$ | the set of all discrete logistics nodes in the network |
| $R$ | the set of transportation routes between logistics nodes, $R \subseteq N \times N$ |
| $i, j$ | the set of different logistics nodes in the network, $i, j \in N$ |
| $k, m$ | the set of hub points in the network, $k, m \in N$ |
| **Parameters** | |
| $p$ | the number of hub points to be built |
| $c_{ij}^{A(B)}$ | the unit transportation cost of enterprise A(B) from node $i$ to node $j$ |
| $\alpha_{km}$ | the coefficient of scale economy transporting between hub points $k$ and $m$ |
| $f_k^{A(B)}$ | the fixed cost to construct the hub point $k$ by enterprise A(B) |
| $W_{ij}$ | the total potential market logistics service demand from node $i$ to node $j$ |
| $\beta_{ij}$ | the price elasticity of demand coefficient of the market from node $i$ to node $j$ |
| $\gamma_{ij}$ | the substitutability coefficient of demand to price from node $i$ to node $j$ |
| $S_{ij}^{A(B)}$ | the processing capacity of enterprise A(B) for the market demand from node $i$ to node $j$ |
| $K^{A(B)}$ | the total carbon credits for enterprise A(B) |
| $e^{A(B)}$ | the average amount of carbon credits required by enterprise A(B) to provide one unit of service |
| $TP_{ij}$ | the unit price charged for transferring excess capacity between enterprises A and B from node $i$ to node $j$ |
| $w$ | the unit price charged for transferring carbon credits between enterprises A and B |

### 3.2. Co-Opetitive Scenario

In a duopoly enterprise market, when the service capacity (transport capacity, carbon credits, etc.) of one enterprise is insufficient and the service capacity of the other enterprise is surplus, the enterprise capacity is allowed to be transferred between the two enterprises. The hub-and-spoke logistics network under the competing relationship is jointly constructed by the duopoly enterprises, the number of hub points $p$ is known, while there is no limit on the processing capacity of the hub points. The following is a brief introduction to the settings of the objective functions and constraints.

The objective of this section is to maximize the profit of the duopoly enterprises in the market at the same time, specifically considering the profit from the provision of logistics services, the cost of inter-enterprise capacity transfer (or generate revenue), and the fixed cost of hub construction.

For the first part of the objective function, this paper considers the profit obtained from logistics service as the product of the difference between the charge price and unit cost of providing a unit of service and the total amount of service provided. The specific is as follows,

$$
\begin{aligned}
profit_A &= \left[ P_{ij}^A - \left( c_{ik}^A + \alpha_{km} c_{km}^A + c_{jm}^A \right) \right] \cdot D_{ij}^A y_{ik}^A y_{jm}^A, \\
profit_B &= \left[ P_{ij}^B - \left( c_{ik}^B + \alpha_{km} c_{km}^B + c_{jm}^B \right) \right] \cdot D_{ij}^B y_{ik}^B y_{jm}^B,
\end{aligned}
\tag{1}
$$



in which $\left( c_{ik}^{A} + \alpha_{km} c_{km}^{A} + c_{jm}^{A} \right)$ is the unit price charged by node $i$ through hubs $k, m$ to node $j$ (i.e., $i \to k \to m \to j$). Then, $y_{ik}^{A} y_{jm}^{A}$ denotes the route selection from node $i$ to node $j$, when the value is 1, the route from node $i$ to node $j$ is selected, similarly for enterprise B.

For the second part of the objective function, considering the cost (or revenue) of capacity transfer between enterprises, the costs and expenses generated by capacity transfer are expressed as the product of unit transfer charge price and transfer amount, expressed as $TP_{ij} ta_{ij}$, and carbon credit transfer is similarly expressed as $wE$ [43]. The positive and negative values calculated by the final model represent the revenue and cost of the enterprise. For example, when $ta_{ij} \geq 0$, it means that enterprise B accepts the capacity transfer of enterprise A, and when $ta_{ij} \leq 0$, it is the opposite. The positive and negative meanings of $E$ are the same. For the third part, the fixed costs of enterprises A and B to build the hub points under their responsibility are denoted as $f_{k}^{A} Y_{k}^{A}$ and $f_{k}^{B} Y_{k}^{B}$.

What is more, considering the restrain of demand, it should be noted that when the price decreases, price-sensitive customers tend to buy more. At the same time, as the price of alternatives falls, price-sensitive customers will switch to alternatives. Therefore, in this research the demand for enterprise A and enterprise B is expressed as functions of enterprise A's price $P_{ij}^{A}$ and enterprise B's price $P_{ij}^{B}$ as follows,

$$
\begin{aligned}
D_{ij}^{A} &= W_{ij} - \beta_{ij} P_{ij}^{A} + \gamma_{ij} P_{ij}^{B}, \quad \forall i, j \in N, \\
D_{ij}^{B} &= W_{ij} - \beta_{ij} P_{ij}^{B} + \gamma_{ij} P_{ij}^{A}, \quad \forall i, j \in N.
\end{aligned}
\tag{2}
$$

In general, the substitutability coefficient of demand to price $\gamma_{ij}$ is smaller than the price elasticity of demand $\beta_{ij}$ in Equation (2), and the difference between them can be used to measure the intensity of price competition between enterprises A and B. The greater the difference, the less obvious the competition.

Based on the above description, the bi-objective mixed integer non-linear programming model is constructed as follows,

In the objective function, the first part denotes the profit earned by enterprises A and B for providing logistics services, respectively. The second part denotes the revenue generated or cost consumed by the transfer of residual transportation capacity and carbon credits between enterprises. The third part denotes the fixed costs of enterprises A and B to build the hub points under their responsibility. Thus, the overall represents the profit earned by enterprises A and B operating in the market.

Constraints (*a*1) and (*a*2) indicate that the capacity limit of service demand that enterprises A and B can meet should be between 0 and their own capacity plus (or minus) transferred capacity. Constraints (*a*3) and (*a*4) indicate that the total service demand provided by the two enterprises should also meet the limit of the corresponding carbon credits. Constraints (*a*5) and (*a*6) denote the price elasticity of demand and the substitution effect of demand on price in the market. Constraint (*a*7) denotes that the volume of logistics services provided by enterprises A and B as a whole cannot exceed the potential total demand in the market. Constraint (*a*8) denotes that either hub point is built by only one of the two enterprises A and B. Constraint (*a*9) denotes that the two enterprises build a total of $p$ hubs in the network. Constraints (*a*10) and (*a*11) denote a single assignment of non-hub points to hub points. Constraints (*a*12)–(*a*15) denote that non-hub points can only be assigned to open hub points.

$$
\left\{
\begin{aligned}
& \max \sum_{i,j,k,m\in N}\left[P_{ij}^A - \left(c_{ik}^A + \alpha_{km}c_{km}^A + c_{jm}^A\right)\right]\cdot D_{ij}^A y_{ik}^A y_{jm}^A + \sum_{i,j\in N} TP_{ij}ta_{ij} + wE - \sum_{k\in N} f_k^A Y_k^A \\[2mm]
& \max \sum_{i,j,k,m\in N}\left[P_{ij}^B - \left(c_{ik}^B + \alpha_{km}c_{km}^B + c_{jm}^B\right)\right]\cdot D_{ij}^A y_{ik}^B y_{jm}^B - \sum_{i,j\in N} TP_{ij}ta_{ij} - wE - \sum_{k\in N} f_k^B Y_k^B \\[2mm]
& \text{subject to:} \\
& \quad 0 \le D_{ij}^A \le S_{ij}^A - ta_{ij}, \quad \forall i,j\in N && (a1) \\[1mm]
& \quad 0 \le D_{ij}^B \le S_{ij}^B + ta_{ij}, \quad \forall i,j\in N && (a2) \\[1mm]
& \quad 0 \le e^A \sum_{i,j\in N} D_{ij}^A \le K^A - E && (a3) \\[1mm]
& \quad 0 \le e^B \sum_{i,j\in N} D_{ij}^B \le K^B + E && (a4) \\[1mm]
& \quad D_{ij}^A = W_{ij} - \beta_{ij}P_{ij}^A + \gamma_{ij}P_{ij}^B, \quad \forall i,j\in N && (a5) \\[1mm]
& \quad D_{ij}^B = W_{ij} - \beta_{ij}P_{ij}^B + \gamma_{ij}P_{ij}^A, \quad \forall i,j\in N && (a6) \\[1mm]
& \quad D_{ij}^A + D_{ij}^B \le W_{ij}, \quad \forall i,j\in N && (a7) \\[1mm]
& \quad Y_k^A + Y_k^B \le 1, \quad \forall k\in N && (a8) \\[1mm]
& \quad \sum_{k\in N}(Y_k^A + Y_k^B) = p && (a9) \\[1mm]
& \quad \sum_{k\in N} y_{ik}^A(Y_k^A + Y_k^B) = 1, \quad \forall i\in N && (a10) \\[1mm]
& \quad \sum_{k\in N} y_{ik}^B(Y_k^A + Y_k^B) = 1, \quad \forall i\in N && (a11) \\[1mm]
& \quad \sum_{i\in N} y_{ik}^A \ge Y_k^A + Y_k^B, \quad \forall k\in N && (a12) \\[1mm]
& \quad Y_k^A + Y_k^B \ge y_{ik}^A, \quad \forall i,k\in N && (a13) \\[1mm]
& \quad \sum_{i\in N} y_{ik}^B \ge Y_k^A + Y_k^B, \quad \forall k\in N && (a14) \\[1mm]
& \quad Y_k^A + Y_k^B \ge y_{ik}^B, \quad \forall i,k\in N && (a15) \\[1mm]
& \quad Y_k^A, Y_k^B \in \{0,1\}, \quad \forall k\in N && (a16) \\[1mm]
& \quad y_{ik}^A, y_{ik}^B, y_{jm}^A, y_{jm}^B \in \{0,1\}, \quad \forall i,j,k,m\in N. && (a17)
\end{aligned}
\right.
\tag{3}
$$

### 3.3. Complete Competitive Scenario

Different from the co-opetition scenario, in the perfect competition scenario, enterprise A and enterprise B build their own required hubs separately. Enterprise A builds hubs in the network with $k_A$, $m_A$, while enterprise B builds hubs in the network with $k_B$, $m_B$. $\alpha_{k_A m_A}$, $\alpha_{k_B m_B}$ represents the coefficient of scale economy transporting between $k_A$ and $m_A$, $k_B$ and $m_B$, respectively. $Y_{k_B}$ represents that in complete competition scenario, whether the hub point is built by enterprise A or B. If the location is chosen consistently by enterprises A and B, assuming that this location point can accommodate the next two hubs; and the number of hubs set by enterprises A and B in the hub-and-spoke logistics network is known as $p$ and $r$, respectively. The bi-objective mixed integer non-linear programming model is constructed as follows,

$$
\left\{
\begin{aligned}
&\max \sum_{i,j,k_A,m_A \in N} \left[ P_{ij}^A - \left( c_{ik_A} + \alpha_{k_A m_A} c_{k_A m_A} + c_{jm_A} \right) \right] \cdot D_{ij}^A y_{ik_A} y_{jm_A} - \sum_{k_A \in N} f_{k_A} Y_{k_A} \\
&\max \sum_{i,j,k_B,m_B \in N} \left[ P_{ij}^B - \left( c_{ik_B} + \alpha_{k_B m_B} c_{k_B m_B} + c_{jm_B} \right) \right] \cdot D_{ij}^B y_{ik_B} y_{jm_B} - \sum_{k_B \in N} f_{k_B} Y_{k_B}
\end{aligned}
\right.
$$

subject to:

$$0 \le D_{ij}^A \le S_{ij}^A, \quad \forall i,j \in N \qquad (b1)$$

$$0 \le D_{ij}^B \le S_{ij}^B, \quad \forall i,j \in N \qquad (b2)$$

$$0 \le e^A \sum_{i,j \in N} D_{ij}^A \le K^A \qquad (b3)$$

$$0 \le e^B \sum_{i,j \in N} D_{ij}^B \le K^B \qquad (b4)$$

Constraints $(a5) - (a7)$

$$\sum_{k_A \in N} Y_{k_A} = p \qquad (b5)$$

$$\sum_{k_B \in N} Y_{k_B} = r \qquad (b6)$$

$$\sum_{k_A \in N} y_{ik_A} Y_{k_A} = 1, \quad \forall i \in N \qquad (b7)$$

$$\sum_{k_B \in N} y_{ik_B} Y_{k_B} = 1, \quad \forall i \in N \qquad (b8)$$

$$\sum_{i \in N} y_{ik_A} \ge Y_{k_A}, \quad \forall k_A \in N \qquad (b9)$$

$$Y_{k_A} \ge y_{ik_A}, \quad \forall i,k_A \in N \qquad (b10)$$

$$\sum_{i \in N} y_{ik_B} \ge Y_{k_B}, \quad \forall k_B \in N \qquad (b11)$$

$$Y_{k_B} \ge y_{ik_B}, \quad \forall i,k_B \in N \qquad (b12)$$

$$Y_{k_A}, Y_{k_B} \in \{0,1\}, \quad \forall k_A, k_B \in N \qquad (b13)$$

$$y_{ik_A}, y_{ik_B}, y_{jm_A}, y_{jm_B} \in \{0,1\}, \quad \forall i,j,k_A,k_B,m_A,m_B \in N. \qquad (b14)$$

(4)

Similar to the co-opetitive scenario, in the objective function, the first part represents the profits earned by enterprises A and B in providing logistics services, respectively. The second part represents the fixed costs to be spent by enterprises A and B in building hubs, which need to be reduced from the profits earned in providing logistics services. Therefore, the function as a whole represents the profit earned by enterprise A (or B) operating in the market under a complete competition scenario.

Constraints ($b1$) and ($b2$) denote capacity constraints on the service demand that the enterprise can satisfy. Constraints ($b3$)–($b4$) restrain on carbon credits and the price demand function. Constraints ($b5$)–($b14$) are related to the construction of the hub-and-spoke logistics network, which include the quantity of hub points, the single assignment of non-hub points to hub points, and the fact that non-hub points can only be assigned to open hub points.

### 3.4. Perfect Cooperative Scenario

Based on the assumption of a cooperative relationship, the two parties of the duopoly enterprises enter into a relevant framework agreement with the overall profit maximization as the objective, that is, while agreeing to set a uniform market price, then the price cannot be adjusted without authorization after the agreement. The index set, parameter setting,

and variables in the perfect cooperation scenario are completely consistent with those in the co-opetition scenario. The single-objective mixed-integer non-linear programming model is constructed as follows,

$$
\begin{cases}
\max \sum_{i,j,k,m \in N} \left[ P_{ij}^A - \left( c_{ik}^A + \alpha_{km} c_{km}^A + c_{jm}^A \right) \right] \cdot D_{ij}^A y_{ik}^A y_{jm}^A - \sum_{k \in N} f_k^A Y_k^A + \\
\qquad \sum_{i,j,k,m \in N} \left[ P_{ij}^B - \left( c_{ik}^B + \alpha_{km} c_{km}^B + c_{jm}^B \right) \right] \cdot D_{ij}^B y_{ik}^B y_{jm}^B - \sum_{k \in N} f_k^B Y_k^B \\
\text{subject to:} \\
\quad \text{Constraints } (a1) - (a4), (a8) - (a17) \\
\quad D_{ij}^A = W_{ij} - \beta_{ij} P_{ij}^A, \quad \forall i,j \in N \qquad (c1) \\
\quad D_{ij}^B = W_{ij} - \beta_{ij} P_{ij}^B, \quad \forall i,j \in N \qquad (c2) \\
\quad D_{ij}^A + D_{ij}^B \le W_{ij}, \quad \forall i,j \in N \qquad (c3) \\
\quad P_{ij}^A = P_{ij}^B, \quad \forall i,j \in N. \qquad (c4)
\end{cases}
\tag{5}
$$

In the composition of the objective function, the first part denotes the profit earned by enterprises A and B for providing logistics services, which is similar to the scenario of the co-opetition; the second part denotes the fixed cost $f_k^A Y_k^A$ and $f_k^B Y_k^B$ spent by enterprises A and B for building the hub point, and the hub-and-spoke logistics network is performed jointly by A and B. The fixed costs are reduced from the profits earned by providing logistics services. Thus, the overall represents the profit earned by enterprises A and B operating in the market in a perfect cooperative scenario. It should be noted that the transfer of surplus capacity between enterprises A and B is treated as an internalized transfer, and the benefits generated or costs consumed are eliminated and are no longer reflected in the objective value function.

Constraint ($c3$) represents the same meaning as constraint ($a7$) in the scenario of the co-opetition. Except that constraints ($c1$) and ($c2$) denote the price demand function and constraint ($c4$) denotes the common pricing of the corporate strategic framework agreement.

### 3.5. Model Solution

In order to solve the result of market equilibrium in a competitive scenario, Rohaninejad et al. [16] pointed out that when the profit in competition deviates minimally from the optimal profit, each enterprise will no longer be willing to change its decision, at which point the optimal solution is the result of the equilibrium state.

In the co-opetition, the purpose of cooperation on co-built hubs and capacity transfer is to reduce fixed costs and increase profits of the two enterprises directly or indirectly through the transfer of surplus capacity, both of which will have no impact on demand-price competition. The final solution of the model is still the "equilibrium" state where both parties want to maximize profits. Therefore, the idea of solving the bi-objective model in a co-opetitive scenario is similar to that of complete competition, and the solution models in both scenarios are as follows,

$$
\min \frac{profit_A^* - profit_A^1}{profit_A^*} + \frac{profit_B^* - profit_B^1}{profit_B^*},
$$

where $profit_A^1$ represents the profit of enterprise A in complete competitive or co-opetitive scenario, and $profit_B^1$ represents a similar meaning, while $profit_A^*$ and $profit_B^*$ can be solved according to model (6). Essentially, this bi-objective to single-objective transformation utilizes the idea of weighted sums with weights $\frac{1}{profit_A^*}$ and $\frac{1}{profit_B^*}$, respectively. Accordingly, the two bi-objective models under the co-opetitive and complete competitive scenario are transformed into a single-objective model.

Taking enterprise A as an example, the optimal profit model to get $profit_A^*$ is as follows,

$$
\left\{
\begin{array}{ll}
\max \displaystyle\sum_{i,j,k_A,m_A \in N} \left[ P_{ij}^A - \left( c_{ik_A} + \alpha_{k_A m_A} c_{k_A m_A} + c_{jm_A} \right) \right] * D_{ij}^A y_{ik_A} y_{jm_A} - \displaystyle\sum_{k_A \in N} f_{k_A} Y_{k_A} & \\[2mm]
\text{subject to:} & \\[2mm]
\quad 0 \leq D_{ij}^A \leq S_{ij}^A, \quad \forall i,j \in N & (d1) \\[2mm]
\quad 0 \leq e^A \displaystyle\sum_{i,j \in N} D_{ij}^A \leq K^A & (d2) \\[2mm]
\quad D_{ij}^A = W_{ij} - \beta_{ij} P_{ij}^A, \quad \forall i,j \in N & (d3) \\[2mm]
\quad D_{ij}^A \leq W_{ij}, \quad \forall i,j \in N & (d4) \\[2mm]
\quad \displaystyle\sum_{k_A \in N} Y_{k_A} = p & (d5) \\[2mm]
\quad \displaystyle\sum_{k_A \in N} y_{ik_A} Y_{k_A} = 1, \quad \forall i \in N & (d6) \\[2mm]
\quad \displaystyle\sum_{i \in N} y_{ik_A} \geq Y_{k_A}, \quad \forall k_A \in N & (d7) \\[2mm]
\quad Y_{k_A} \geq y_{ik_A}, \quad \forall i, k_A \in N & (d8) \\[2mm]
\quad Y_{ik_A} \in \{0,1\}, \quad \forall k_A \in N & (d9) \\[2mm]
\quad y_{ik_A}, y_{jm_A} \in \{0,1\}, \quad \forall i,j,k_A,m_A \in N. & (d10)
\end{array}
\right. \tag{6}
$$

In the objective function, the first part represents the profit that enterprise A makes by providing logistics services, and the second part represents the fixed cost that the enterprise needs to spend to build the hub point. Thus, the overall represents the optimal profit that enterprise A can obtain in the market based on its own ability.

Constraints ($d1$) and ($d2$) denote the enterprise capacity and carbon credit constraints, respectively. Constrains ($d3$) and ($d4$) restrain the price demand function and the potential aggregate market demand. Constraints ($d5$)–($d10$) are related to the construction of the hub-and-spoke logistics network. Similarly, the model for calculating the optimal profit $profit_B^*$ for enterprise B is similar to that for enterprise A.

## 4. Numerical Analysis

In this section, results are displayed and analyzed in four different scenarios through small data from four dimensions. After that, sensitivity analysis is performed using complete CAB(with 25 nodes) and TR (with 81 nodes) datasets for the co-opetition scenario.

### 4.1. Parameter Setting

Referring to Mahmoodjanloo et al. [33] on price parameters considered for pricing, the settings of price elasticity of demand and substitution effect coefficients need to consider the intensity of competition between the two enterprises. When studying the entry of new enterprises into the market, if there is no price competition in the market, the price elasticity of demand $\beta$ is 20; if there is price competition in the market, the price elasticity of demand $\beta$ is 30, and the substitution effect coefficient $\gamma$ is 6. The different settings of the price elasticity coefficients considered the fact that when there is price competition in the market, consumers' sensitivity to the price of logistics services will increase, and some customers will transfer between enterprises.

In addition, the fixed cost of hub point investment and construction is set to be arbitrarily chosen from the interval [450,000, 550,000] [44] , and the coefficient of scale economy $\alpha$ for transportation between hub points is 0.6. Considering the general case of the linear relationship between transportation distance and transportation cost, here we use the ratio of transportation distance to a constant (say 20) to represent the unit

distance transportation cost $c_{ij}$ between nodes [5]. These settings take into account the ratio between transportation cost and hub cost, and balance the cost of hub construction as much as possible.

For the inter-node market potential demand $W_{ij}$, the traffic data in the standard dataset are used. The inter-node transportation capacity $S_{ij}$ that can be served by the duopolistic enterprises is generated as follows. It is assumed that enterprise A is more capable than enterprise B. First obtain a random set of data $S_{ij}$ from the interval $[\frac{1}{3}W_{ij}, \frac{2}{3}W_{ij}]$, then calculate $W_{ij} - S_{ij}$, and finally the larger value of the two is chosen for the capacity of enterprise A, and the other value is the capacity of enterprise B. The carbon credit $e$ required to provide one unit of demand is set equally to 1 for both A and B. The total carbon credit $K$ is generated using the total transportation capacity of enterprises as the reference value. In addition, regarding the residual capacity transfer charge price $TP_{ij}$ in the co-opetitive and perfect cooperative scenarios as $0.2*$ the pricing of enterprise A in the complete competitive scenario, the carbon credit transfer price $w$ is set in a similar way as the residual capacity transfer price.

### 4.2. Numerical Analysis Based on the Small Dataset

The small-scale experiment refers to the study of Čvokić and Stanimirović [35], experimenting with the CAB dataset while considering pricing and single allocation at a non-pivotal point with the goal of profit maximization. The 10 nodes randomly selected from the CAB dataset include 4 (Chicago), 5 (Cincinnati), 7 (Dallas/Fort Worth), 8 (Denver), 9 (Detroit), 11 (Kansas City), and 17 (New York), 19 (Phoenix), 21 (St. Louis), and 23 (Seattle), in which five of the randomly selected logistics nodes are near the national boundary, and five cities are in the interior (see Figure 2).

#### 4.2.1. The Design of "Channel + Hub + Network" Logistics System

The final locations of the hub points in different scenarios and the allocation of other cities to the hub cities are shown in Figure 2. For the two enterprises, in the complete competition scenario, cities 4 and 8 (Figure 2a) and cities 4 and 17 (Figure 2b) are selected as hub points, respectively. In the co-opetition and perfect cooperation scenarios, cities 4 and 9 (Figure 2c,d) are selected as hub points for both, but with different allocation. The same results as in the article [35] further illustrate the validity of hub point selection.

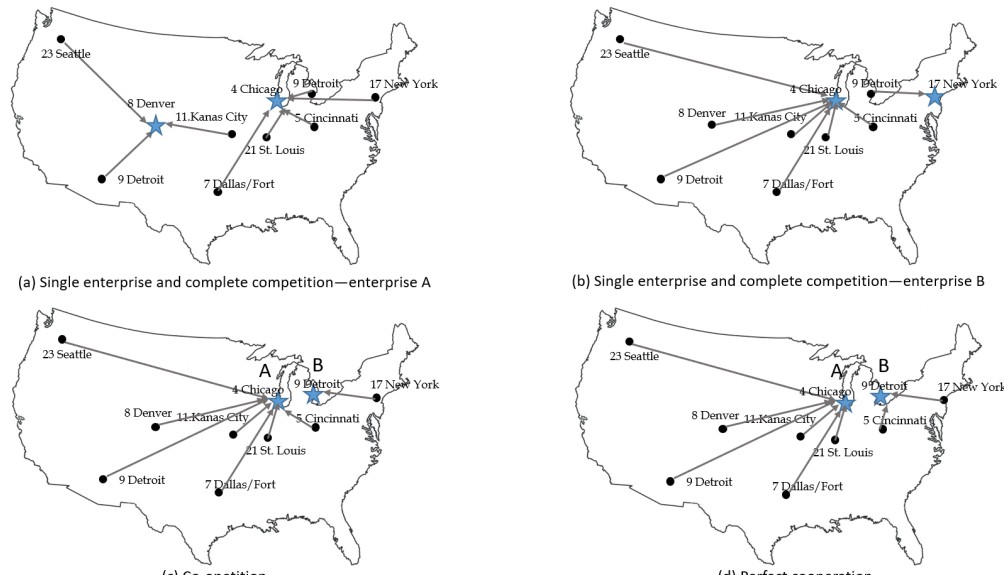

**Figure 2.** Design of four contextual hub-spoke logistics networks based on a small dataset.

As for the hub-and-spoke logistics network, from Figure 2, the results of each of the two enterprises are the same for the single enterprise and the complete competition scenarios. The reason for this is that the relevant cost factors considered by the enterprises when laying out the hub-and-spoke logistics network in both scenarios have not yet changed, and the change in the amount of services provided by the enterprises between the points is based on the total market demand. The overall distribution of demand in the network is similar. On the other hand, there are differences in the selection of hubs and the distribution of non-hubs in the four scenarios, especially when strategic cooperation between enterprises is involved or not. Thus, in the market operating environment, it is crucial for enterprises to make strategic choices for themselves in terms of competition and cooperation.

4.2.2. Business Operating Profit

From the perspective of business profits, according to Table 2, as the degree of cooperation increases, the profits of enterprise A, enterprise B and total profits in the complete competitive, co-opetitive, and perfect cooperative scenarios all increase sequentially. For enterprise A, even the profit in the perfect cooperative scenario has exceeded the profit in the single enterprise scenario in the same market demand environment.

**Table 2.** Realistic operating profits of companies in four scenarios based on small dataset.

|  | Single Enterprise | Complete Competition | Co-Opetition | Perfect Cooperation |
|---|---|---|---|---|
| Profit of A | 1,188,524,596 | 996,599,244 | 1,010,784,330 | 1,214,740,610 |
| Profit of B | 1,157,405,364 | 939,251,688 | 952,460,802 | 1,154,259,677 |
| Total profits |  | 1,935,850,933 | 1,963,245,131 | 2,369,000,287 |

As mentioned above, it is assumed that enterprise A is more capable than enterprise B. In the perfect cooperative scenario, the profit of enterprise B in the perfect cooperative scenario has not yet exceeded that of the single enterprise scenario because enterprise B has weaker capacity and profits by setting high prices in the single enterprise scenario, but its pricing is reduced in the perfect cooperative scenario considering system profit maximization, and it needs to pay certain costs for the accepted capacity transfer from enterprise A, which makes profits less. Even though the profit has reduced, the decrease is small. Overall, the perfect cooperative scenario is the most profitable when there are two enterprises in the market.

4.2.3. Enterprise Service Pricing

This paper introduces a customer demand model with price elasticity and demand substitution effect to consider service pricing. Figure 3 depicts the pricing of different routes and the corresponding customer service demand based on price for enterprise A and enterprise B in four scenarios, where "R4 5" represents the route from node 4 to node 5, and the same for the others.

Looking at the pricing of different routes as a whole, most routes are priced within the range (0, 300), with a few routes exceeding this range, such as R4 17, because when the price sensitivity of customers is the same across the market, the duopoly enterprises are able to capture the demand of the corresponding market without setting low prices for routes with high potential customer service demand in the market segment. At the same time, it can be seen from the figure that for some routes, enterprises do not provide services. Taking R5 19 as an example, further exploration reveals that the potential market demand on this route is less, down to only 1041. However, the cost of this route is relatively high, which has reached 78.54. That is, in the case of consistent customer sensitivity in all market segments, both enterprises choose to give up this part of the route for the profit maximization goal.

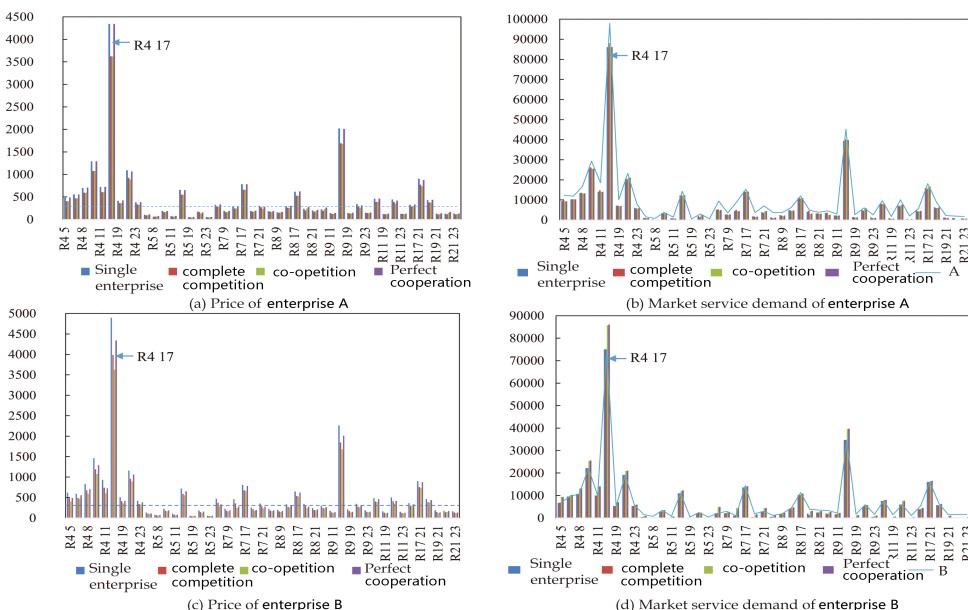

**Figure 3.** Comparison chart of enterprise service pricing for four scenarios based on a small dataset.

Overall, the price of enterprise B is higher than that of enterprise A in the single enterprise and complete competitive scenario, and slightly higher for most routes in the co-opetition, due to enterprise B's limited capacity and carbon credits. In particular, for enterprise B, it does not offer service even in less profitable route markets, such as R21 23, due to its carbon credit limitations. The opposite is true for enterprise A. From the comparison of service pricing in the four scenarios (Figure 3a,c), the market price in the co-opetition is lower than that in the single enterprise and perfect cooperative scenarios, so that in a market where demand is determined by price, firm price competition is more favorable for customers to pay less for logistics services. At the customer level, the price competition is more favorable for customers to pay less for logistics services, the co-opetition and complete competition are the preferred modes of market operation for customers.

4.2.4. Market Equilibrium State

Based on the analysis of enterprise service pricing, the market equilibrium state solved in this paper is further analyzed in this section. Take the equilibrium states of enterprises A and B with complete competitive scenario route R4 5 and co-opetition route R5 7 as examples, their results are shown in Figure 4. For the Figure 4a, in the complete competitive scenario of the R4 5 market segment, when the decision of enterprise A is unchanged, enterprise B can only change the equilibrium by raising its own price due to capacity constraints; when the decision of enterprise B is unchanged, enterprise A can increase its customer service by lowering its price, so the model solution results in this situation are expressed in the upper left region. The results of other regions are derived by raising the price of enterprise B or lowering the price of enterprise A. For the Figure 4b, in the R5 7 segment of co-opetition, enterprises A and B have almost the same capacity and both have surplus, and there is room for price reduction, so the model solution results in this scenario are expressed in the upper right region, and the other regions are derived by reducing the price of enterprise A or enterprise B to calculate profits.

Specifically, the vertical arrows in Figure 4 represent the low to high prices of enterprise A and the horizontal arrows represent the low to high prices of enterprise B. The sizes of circles represent the service capacity of enterprises. The larger the circle, the more capable the enterprise. The circle with outer ring represents the enterprise's own better profit choice under the current decision of the other enterprise. It should be noted that the high and low prices are relative concepts. For example, when one of the two rectangles at the top of the graph represents the profit of enterprise B, the point with an outer ring means that

when enterprise A's decision is high price, enterprise B's better profit choice is low price (see Figure 4a, top left corner, for the complete competitive scenario) or high price (see Figure 4b, top right corner, for the co-opetition scenario).

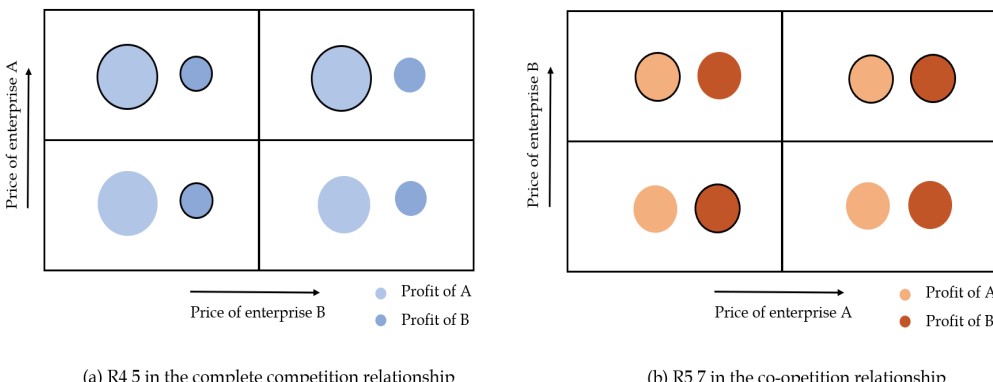

(a) R4 5 in the complete competition relationship

(b) R5 7 in the co-opetition relationship

**Figure 4.** Diagram of market equilibrium under the cooperative competition.

As seen in Figure 4a, for R4 5 segment of complete competition scenario, no matter enterprise A takes what kind of price, enterprise B chooses the low price profit better, while the opposite is true for enterprise A. Only when enterprises A and B are in the upper-left corner (high price, low price) can both of them maximize their profits under each other's strategies and reach market equilibrium. In this state, any change in the profit strategy of one enterprise will cause its own profit to deviate from the better outcome. Therefore, the upper left region is the equilibrium representation of the model solution result. Similar results can be obtained using the same analysis in Figure 4b for the co-opetition of R5 7 market segment.

Through the specific analysis from four aspects of hub-and-spoke logistics network design, operating profit, service pricing and equilibrium state, we come to the conclusion that, on the whole, the co-opetition scenario is better than the other three scenarios. The co-opetition scenario realizes easier than the perfect cooperation scenario and has higher profits than the complete competition scenario. Based on this, sensitive analysis is carried out for the co-opetition scenario in the next section.

### 4.3. Sensitivity Analysis

This section further develops the sensitivity analysis using the complete CAB and TR datasets in the context of coexistence of competition and cooperation. The sensitivity analysis on the different parameters is carried out with "coefficient of scale economy $\alpha$ of 0.6, number of hub points $p$ of 3 (CAB dataset) and 5 (TR dataset), transfer price $TP$ of 0.2* benchmark, enterprise A(B) capacity $S_{ij}^{A(B)}$ and carbon credits $K^{A(B)}$ are both 1.4" as the benchmark for the expansion.

4.3.1. Economies of Scale Transportation Coefficients and Number of Hub Points

Taking the CAB dataset as an example, Figure 5 summarizes the number of pivot points $p$ taking the value 3. From the selection of hub points, it can be seen that with the number of hub points determined, the location of hub points becomes more concentrated as the scale transportation factor $\alpha$ increases.

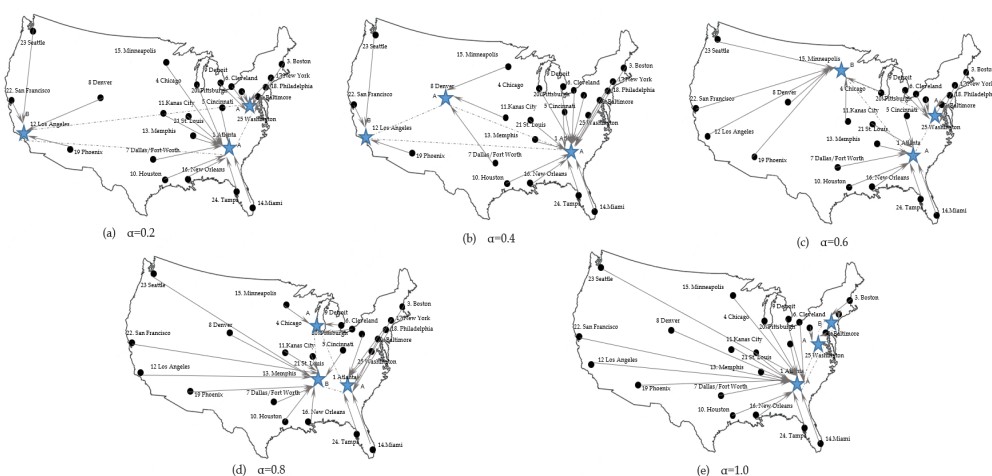

**Figure 5.** Hub-and-spoke logistics network design with different coefficient of scale economy.

As shown in the Figure 5, when α is 0.2, the selected hubs are cities 1, 12, and 25, which are the most dispersed; when α is 1.0, i.e., there are no economies of scale in transportation between hubs, the selected hubs are cities 1, 18, and 12. That is to say, when the effect of scale economy is more obvious, long-distance nodes through the hub point transport can effectively reduce the impact of the distance, while the nodes closer to the hub point can be directly assigned to the hub point to further reduce transport costs. On the other hand, when the effect of scale economy is not obvious or there is no effect of scale economy, the network tends to choose the closer point to minimize the cost of trunk transport.

Furthermore, in Figure 6, it can be seen from the first column that when other parameters are held constant, as the coefficient of scale economy α increase, the decrease in profit and total profit of duopoly enterprises A and B is less sensitive to the change of the coefficient of scale economy. This is due to the fact that when the coefficient is smaller, the cumulative value of transportation costs between hub points will shrink to a greater extent leading to a greater value of increased profits.

From the second column of Figure 6, when other parameters are constant, as the number of fixed hub points constructed increases, the raise in the profit of the duopoly enterprises A, B and the total profit shows a gradual decrease, reflecting that though the network transportation costs decrease, the difference value between the savings in transportation costs and the fixed costs required to build hub points becomes smaller.

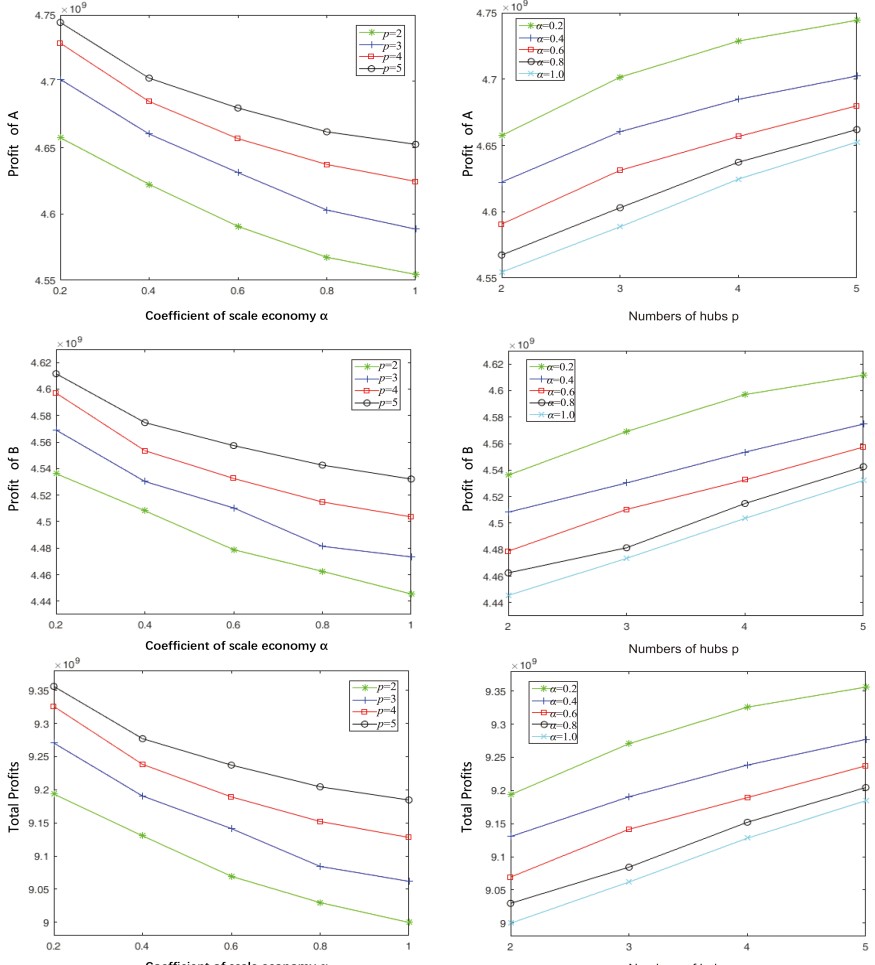

**Figure 6.** Impact of changes in the coefficient of scale economy and the number of hub points on profits based on the CAB dataset.

### 4.3.2. Transfer Price of Residual Capacity

This section studies the effect of the transfer price of residual capacity $TP$ on the profit of enterprises A and B. The transfer price is set with the price of enterprise A under the complete competition scenario as the benchmark. As shown in Figure 7 and Table 3, with the increase in the transfer price of residual capacity, the profit of enterprise A increases and the profit of enterprise B decreases, the profit difference value gradually increases, and the difference value is the smallest at the price of 0.2*basis. Meanwhile, as the transfer price of residual capacity increases, the sensitivity of the increase (enterprise A) or decrease (enterprise B) of enterprise's profit to the change of transfer price gradually decreases, which means when the transfer price is low, a small change in price will have a higher impact on profit, and when the transfer price is high, this effect will no longer be obvious.

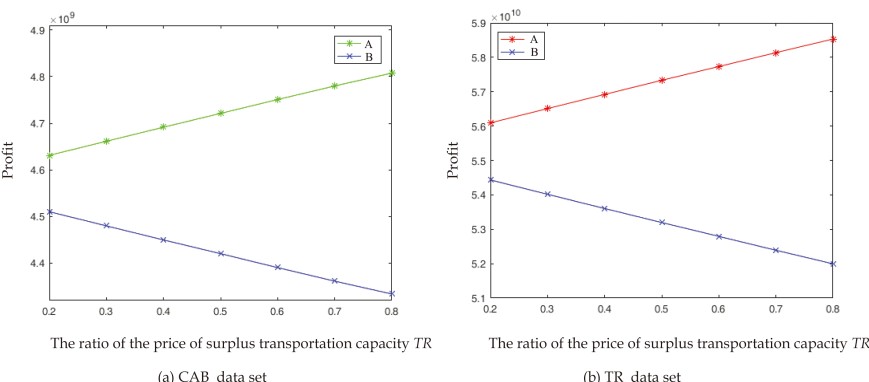

**Figure 7.** Impact of change in the transfer price of residual capacity on profits.

**Table 3.** Change of profit in the case of transfer price of residual capacity variation.

| | CAB Data Set | | TR Data Set | |
| --- | --- | --- | --- | --- |
| **Change of TP Ratio** | **Enterprise A** | **Enterprise B** | **Enterprise A** | **Enterprise B** |
| 0.2→0.3 | 30,245,554 | −30,301,796 | 417,601,044 | −417,645,294 |
| 0.3→0.4 | 30,176,995 | −30,137,818 | 412,257,624 | −412,355,734 |
| 0.4→0.5 | 29,514,826 | −29,513,978 | 408,624,209 | −408,599,163 |
| 0.5→0.6 | 29,826,193 | −29,815,350 | 401,630,926 | −401,951,268 |
| 0.6→0.7 | 29,112,959 | −29,119,598 | 403,923,408 | −403,590,268 |
| 0.7→0.8 | 27,465,387 | −27,571,262 | 395,142,708 | −395,421,491 |

### 4.3.3. Corporate Capacity and Carbon Credits

The different service capacity $S_{ij}^{A(B)}$ and carbon credits $K^{A(B)}$ of enterprise A(B) are crucial for enterprise operation. Thus, Figure 8 explores the impact of different capacity ratio settings of enterprises A and B on enterprise profits.

From Figure 8, the profits of enterprises A and B are almost the same when the ratio of enterprise A to enterprise B's capabilities is set to 1.2 or when both enterprises have the same capabilities. For example, for the CAB dataset, when the ratio is set to 1.2, the profits of A and B are 4,537,769,512 and 4,531,006,049, respectively.

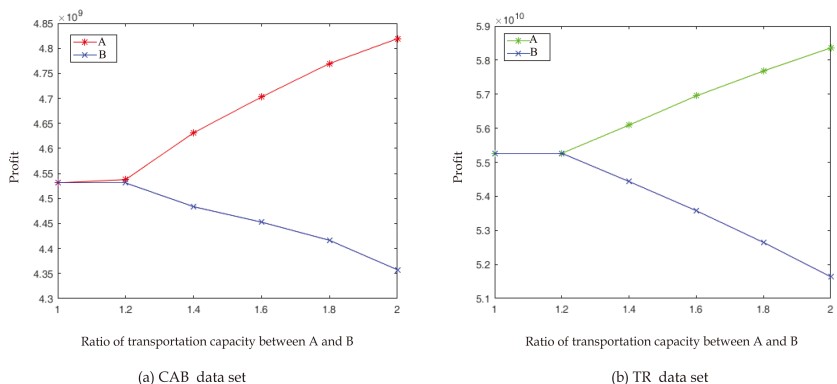

**Figure 8.** Effect of changes in the ratio of the two enterprises' capabilities on profit.

It shows that when capabilities of A and B are similar, the profits made by the enterprises providing logistics services in the market are also almost the same.

When the capacity ratio exceeds 1.2, for enterprise A, as the proportion of capacity increases, the amount of services that can be provided to the market increases, resulting in a corresponding reduction in pricing while declining the transfer price of residual capacity, which leads to the reduction in the sensitivity of profits to changes in the capacity ratio.

For enterprise B, as the capability ratio increases, enterprise B's lack of capability will affect its ability to provide services to the market. When the capacity is lower, the number

of accepted capacity transfer from enterprise A increases, the cost of operating expenses of the enterprise increases accordingly, and the combined effect of these two aspects leads to the reduced value of profit of enterprise B is more sensitive to the change of capacity ratio.

### 4.4. Implications

From the above analysis, we find that enterprises have different operating profits in different situations. Complete cooperation, co-opetition, and perfect competition have successively reduced profits (see Table 2). Complete competition is a common state of existence for enterprises in the market. Perfect cooperation in actual business operations is difficult to achieve, unless the two enterprises are belonging to the same parent enterprise or they are parent-child enterprises. Therefore, the scenario of co-opetition has certain advantages as a scenario with higher profits and easier realization than the scenario of perfect cooperation.

Enterprises can achieve higher profits, more favorable pricing, and higher market service satisfaction rates through strategic cooperation at three levels: transferring their own surplus capacity, carbon credits, and co-building hubs. In particular, when the coexistence of the duopoly enterprises reaches a certain level, the enterprises can try to provide services on certain nodes and routes in the hub-and-spoke logistics network as a pilot, develop small-scale complete cooperation, complete the entire network, and lay the foundation for cooperation.

At the same time, in the co-opetition scenario, the different abilities of duopoly enterprises A and B will have a significant impact on their own profits (see Figure 8). When the enterprises' own capabilities are high, they often take the leading role in strategic cooperation, but enterprises with weak capabilities can still improve its relative status in cooperation by achieving technological innovation, high levels of customer service, more effective problem solutions. On the other hand, enterprises can balance the changes in the profits of the two parties by negotiating transfer price of residual capacity, thereby further promoting long-term strategic cooperation.

In addition, when enterprises are operating in the market, the "price reduction strategy" is not the most effective way for enterprises to stimulate customer demand for logistics services to increase their own profits. In market competition, sometimes price reductions would have a counterproductive effect. For example, in the first quarter of 2020, "price war" of express delivery enterprises in China ultimately failed to achieve the expected operating results. Therefore, when stimulating customer service demand, enterprises should carefully adopt low-price competition strategies and comprehensively evaluate the impact of pricing on business objectives.

For large-scale transportation, the more obvious economic effects of scale transportation can have a greater impact on the increase in corporate profits (see Figure 6). Therefore, enterprises in actual operations can increase the demand for customer service by adopting incentives to implicitly increase the economic effects of scale transportation to gain an increase in profits. However, when the enterprise's incentive measures need to increase the company's other operating costs in other ways, logistics enterprises should carefully choose whether to take such measures. This is because when the economic effects of scale transportation become less obvious, the enterprise's profits reduce. However, this effect will gradually become insignificant as the scale transportation coefficient increases. Therefore, operating enterprises need to weigh the cost of reducing the scale of transportation coefficient and the benefits that it brings to the enterprises, and finally decide on detailed operational measures.

As mentioned above, for logistics enterprises, the construction of facilities at fixed hubs is a long-term measure that will take up a lot of capital flows. Therefore, in actual operations, enterprises tend to focus on long-term strategies to explore when future customer service demands significantly increase, and how much the construction of fixed hubs can help enterprises to save the variable transportation cost. For the construction of fixed hub facilities, as the number of hubs increases, the sensitivity of corporate profits to increase

will become lower (see Figure 6). Therefore, when it is expected that the effect of substantial savings in transportation costs cannot be achieved, enterprises should choose to build fewer fixed hubs and invest in construction later when needed. At the same time, based on the research in this article, logistics enterprises could build a common hub, jointly maintain the operation of the hub, and reduce unnecessary long-term capital occupation by enterprises.

Specifically, for "channel + hub + network" logistics system design, on the one hand, enterprises should try to match their own fixed hub facility construction with the market service demand they need to provide, use higher sensitivity to restrict the investment and construction of non-essential fixed hub facilities, and reduce the occupation of fixed asset investment funds of enterprises due to the large number of pivot points. When the construction of fixed hub facilities does not completely match the demand, enterprises can cooperate with other enterprises to build hubs to match the amount of services that the hub can provide with actual needs. For example, JD Logistics has a large warehouse and distribution center layout across the country. In the early stage, it was only self-operated, and in the later stage, it chose to rent out some warehouses for other enterprises to use in order to increase profits. On the other hand, for logistics enterprises, when a fixed hub has been built in the early stage, the focus of the enterprise's later operations is mainly to stimulate customer demand and hold a good corporate operating system arrangement to maintain a more obvious scale transportation economy, which is conducive to the company's next round of deployment and construction of hub facilities based on profit and loss.

## 5. Conclusions

With the rapid development of the e-commerce industry, people's demand for logistics market has greatly increased. Under these circumstances, there are great opportunities for the logistics enterprises. However, on the other hand, the current market operations of logistics enterprises are also facing challenges. First, the total logistics cost is relatively high, especially the construction of the hub point infrastructure of the hub-and-spoke logistics network requires a lot of capital, the recovery time of the enterprise's fixed capital investment is long, and the wasting phenomenon of resource is serious. Second, when the capacity and resources of an enterprise exceed the market demand for enterprise logistics services, a waste of capacity and resources will occur, such as the capacity of logistics enterprises and the carbon credits of enterprises under the carbon tax policy. Third, the service pricing of logistics enterprises lacks market-oriented pricing thinking. For example, most express enterprises only distinguish between internal and external-province pricing for general services, which lack market rationality.

Based on the above research background, this paper constructs a planning model for the "channel + hub + network" logistics system considering the coexistence of competition and cooperation to meet the challenges above. For the first challenge, this model considers the joint construction of pivot points, which could reduce the capital of the enterprise to construct hub points. For the second challenge, elements like the transfer of enterprises' surplus capacity, carbon credit are also reflected in the model. In this way, the waste is effectively reduced. For the third challenge, this paper introduces demand price elasticity and demand substitution effect to consider service pricing, and then puts forward some suggestions for enterprise pricing strategy.

In addition, after a series of comparative analysis through the benchmark datasets, we find that the operating profits of enterprises in different scenarios are different. In the scenario of co-opetition, the remaining capacity of the enterprise, the transfer of carbon credits, the joint construction and sharing of pivot points, and the common pricing of the price framework agreement in the full cooperation scenario strategic cooperation in other areas are conducive to the improvement of corporate operating profits, better market share and more favorable pricing.

Furthermore, there are still directions that can be improved in the research of this paper. Firstly, consideration of more factors that affect the service demand of logistics enterprises can be joined into future research, such as the service level and the distance from the customer to the service point. Secondly, this paper uses the classic American CAB and Turkish network datasets to solve the model, real data of logistics enterprises in actual operations can be collected for research for verifying. Lastly, in actual enterprise operations, the number of demand points will exceed or even far exceed 81 points, it is feasible to design optimization algorithms for solving larger-scale problems.

**Author Contributions:** Conceptualization J.Z. and H.Z.; Methodology J.Z., K.X. and Y.Z.; Software J.Z. and H.Z.; Validation J.Z. and K.X.; Formal analysis J.Z. and K.X.; Investigation H.Z.; Data curation J.Z., H.Z. and K.X.; Writing–original draft preparation Y.Z.; Writing–review and editing K.X. and J.Z.; Visualization Z.D. and K.X.; Supervision J.Z. All authors have read and agreed to the published version of the manuscript.

**Funding:** This work was supported in part by the grants from the National Natural Science Foundation of China (Grant No. 71872110).

**Institutional Review Board Statement:** Not applicable.

**Informed Consent Statement:** Informed consent was obtained from all subjects involved in the study.

**Data Availability Statement:** The data presented in this study, in anonymized form, are available on request from the authors.

**Acknowledgments:** The authors especially thank the editors and anonymous referees for their kindly review and helpful comments.

**Conflicts of Interest:** The authors declare no conflict of interest.

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
