# Peer review of "Hub-and-Spoke Logistics Network Considering Pricing and Co-Opetition"

_sustainability, doi:10.3390/su13179979_

Round 1

Author Response

Comment #1: In paragraph 4 of Section 2, more literature review on pricing is needed, specifically the literature related to the pricing in a competitive environment.

Answer: Thank you very much for your kind comments. Done!

Following your suggestion, we enriched the literature on pricing in a competitive environment. Please see Lines 124-126 on Page 3 for more details. 

We hope to have fully answered Comment #1.

Comment #2: In Section 3, constrains (a5) and (a6) contain in model (3) and (4), while in model (5) they disappear. What is the reason?

Answer: Thank you very much for your kind comments.

models (3) and (4) represents the substitution effect of demand on price of enterprise A(B). In the complete competition and co-opetition scenarios, there is price competition between enterprises A and B, thus in models (3) and (4), constrains (a5) and (a6) contain . However, in the perfect cooperation scenario, enterprises A and B both agree to a uniform market price. In this case, the substitution effect of demand on price disappear, then  disappear in the model (5).

We hope to have fully answered Comment #2.

Comment #3: In Section 3, the difference between  and  is not clearly explained.

Answer: Thank you very much for your kind comments. Done!

Taking into consideration your feedback, we revised the description to explain the difference between  and , which have been listed as follows:

  • represents that in a complete competition scenario, whether the hub point is built by enterprise A or B.
  • represents that in the perfect cooperation and co-opetition scenarios, whether the hub point is built by enterprise A or B.

Please see Lines 245-246 on Page 8 for more details.

We hope to have fully answered Comment #3.

Comment #4: In the last paragraph of Section 4.2, “Through the specific analysis …than the other three scenarios”. Implications are not very appropriate, which should be modified in the new version.

Answer: Thank you very much for your kind comments. Done!

Taking into consideration your feedback, we revised the description of “Through the specific analysis …than the other three scenarios” as “The co-opetition scenario realizes easier than the perfect cooperation scenario and has higher profits than the complete competition scenario”.

Please see Lines 451-452 on Page 15 for more details.

We hope to have fully answered Comment #4.

Comment #5: The paper requires the following minor corrections, such as:

On page 1 line 21, it should be “demand for”. On page 2 line 60, it should be “construction”. On page 4 line 164, it should be “form”.

Answer: Thank you very much for your kind comments. We have gone through the whole paper, and we polished the language and made several improvements and corrections. Please be aware that in the revised version of our paper, only the major changes have been highlighted.

We hope to have fully answered Comment #5.

Reviewer 2 Report

The present article is based on the perspective of duopoly logistics companies, which together build networks and enable the transfer of excess capacity and carbon credits, and study the design of a hub-and-speak logistics network, which also takes into account the relationship between service prices and co-optation. The article is interesting and provides an important contribution.

Author Response

Comment : The present article is based on the perspective of duopoly logistics companies, which together build networks and enable the transfer of excess capacity and carbon credits, and study the design of a hub-and-speak logistics network, which also takes into account the relationship between service prices and co-optation. The article is interesting and provides an important contribution.

Answer: Thank you very much for your kind comment.

Reviewer 3 Report

The presented article is very interesting and I am expecting new article where you will on concrete examples try to further discuss stated hypotheses.

Author Response

Comment : The presented article is very interesting and I am expecting new article where you will on concrete examples try to further discuss stated hypotheses.

Answer: Thank you very much for your kind comment.

Reviewer 4 Report

The paper is mostly well-written, though it could used some editing, and there are a few instances of awkward phrasing or unclear text. The paper is well organized and thorough and provides an important contribution. Below are a few minor comments for improvement.

Lines 22-23: "In view of the fact that the total logistics cost of the whole society accounts for a significantly higher proportion of GDP than developed countries" This sentence is confusing. "Developed countries" is not a component of GDP.

Lines 28-29: "the situation of exchanging prices for the market in the market price war" It is not clear to me what this means.

Table 1, first variable: "the unit price charged by enterprise A for providing" I think this should say "enterprise A (B)"

Lines 323-324: "the ratio of transportation distance to 20 is used" Is there something missing in this sentence? Is it supposed to say that the ratio of transportation distance to cost is 20?

Does Figure 3 show anything that is not also in Table 2? I don't think they are both necessary, and I think Table 2 is better. I suggest removing Figure 3.

Line 394: I'm not following the meaning of "less for 1041" and "high to 78.54"

Figure 4 is hard to read.  Not all of the routes are labeled, and it would be impossible to do so because of how small the labels would need to be. The columns are so skinny that it is hard to distinguish between them. I'm not sure what we are supposed to learn from this figure other than price and demand is much higher for R4 17. I would think about if there is another way to present this information, or if the figure is even needed. Also, decimal places in the y-axis should be removed.

Figure 5: What do the sizes of the circles represent? I assumed the sizes represented the level of profit, as indicated by the legend, but that does not seem to be the case. Based on the text, it seems that it represents capacity, but the figure does not indicate that. And there is no way of knowing what the outer rings mean or how to interpret the figure without carefully reading the paragraph before it.

Author Response

Comment #1: Lines 22-23: "In view of the fact that the total logistics cost of the whole society accounts for a significantly higher proportion of GDP than developed countries" This sentence is confusing. "Developed countries" is not a component of GDP.

Answer: Thank you very much for your kind suggestion. Done!

The purpose of this paragraph is to explain that the cost spent on logistics takes too much proportion of GDP. Taking into consideration your feedback, we revised the sentence as “In view of the fact that the total logistics cost of the whole society accounts for a significantly high proportion of GDP”.

Please see Lines 22-23 on Page 1 for more details.

We hope to have fully answered Comment #8.

Comment #2: Lines 28-29: "the situation of exchanging prices for the market in the market price war". It is not clear to me what this means.

Answer: Thanks for your kind suggestion.

By using this sentence we mean that in a competitive market, many enterprises choose to lower their prices in order to gain more market share. However, sometimes such strategy would reduce its profit, so enterprises should be cautious to adjust their prices.

Taking into consideration your feedback, we revised the sentence as “the situation of changing prices in the market price war” which might be easier to understand.

Please see Line 28 on Page 1 for more details.

We hope to have fully answered Comment #9.

Comment #3: Table 1, first variable: "the unit price charged by enterprise A for providing" I think this should say "enterprise A (B)".

Answer: Thanks for your valuable comments. Done!

Taking into consideration your feedback, we revised the sentence as follows:

“the unit price charged by enterprise A(B) for providing one unit of logistics service”.

Please see Table 1 on Page 5 for more details.

We hope to have fully answered Comment #10.

Comment #4: Lines 323-324: "the ratio of transportation distance to 20 is used" Is there something missing in this sentence? Is it supposed to say that the ratio of transportation distance to cost is 20?

Answer: Thanks for your instructive comments.

As the relationship between transportation distance and transportation cost is generally linear, we take the unit distance transportation cost cij between nodes as a ratio of transportation distance to a constant, and in this study we set the constant to 20. Taking into consideration your feedback, we revised the sentence as follows:

“Considering the general case of the linear relationship between transportation distance and transportation cost, here we use the ratio of transportation distance to a constant (say 20) to represent the unit distance transportation cost cij between nodes”, which may explain the relationship more clearly.

Please see Lines 324-327 on Page 11 for more details.

We hope to have answered Comment #11.

Comment #5: Does Figure 3 show anything that is not also in Table 2? I don't think they are both necessary, and I think Table 2 is better. I suggest removing Figure 3.

Answer: Thanks for your instructive comments. Done!

Considering that Figure 3 represents the similar meaning as Table 2, Figure 3 has been removed already in the revised version.

We hope to have answered Comment #12.

Comment #6: Line 394: I'm not following the meaning of "less for 1041" and "high to 78.54".

Answer: Thanks for your instructive comments.

In this sentence we mean that in R5 19, the market demand is 1041, and it is low, while its cost is 78.54, which is relatively high. As the original expression may cause confusion, we revised the sentence as follows:

“Taking R5 19 as an example, further exploration reveals that the potential market demand on this route is less, down to only 1041. However, the cost of this route is relatively high, which has reached 78.54.”

Please see Lines 396-398 on Page 13 for more details.

We hope to have answered Comment #13.

Comment #7: Figure 4 is hard to read.  Not all of the routes are labeled, and it would be impossible to do so because of how small the labels would need to be. The columns are so skinny that it is hard to distinguish between them. I'm not sure what we are supposed to learn from this figure other than price and demand is much higher for R4 17. I would think about if there is another way to present this information, or if the figure is even needed. Also, decimal places in the y-axis should be removed.

Answer: Thanks for your instructive comments.

Taking into consideration your feedback, decimal places in the y-axis has been removed. This figure depicts the pricing of different routes and the corresponding customer service demand based on price for enterprise A and enterprise B in four scenarios. Instead of trying to get precise numerical information from the graph, we compare the four different scenarios and observe the trends. For example, looking at the pricing of different routes as a whole, we found that most routes are priced within the range (0, 300). Besides, overall, the price of enterprise B is higher than that of enterprise A in the single enterprise and complete competitive scenario, and slightly higher for most routes in the co-opetition, it is due to enterprise B's limited capacity and carbon credits.

For the above reasons, we believe that readers can still get useful information from this figure. Thus we still keep this figure in the revised version. Thanks again for your kind comments.

We hope to have answered Comment #14.

Comment #8: Figure 5: What do the sizes of the circles represent? I assumed the sizes represented the level of profit, as indicated by the legend, but that does not seem to be the case. Based on the text, it seems that it represents capacity, but the figure does not indicate that. And there is no way of knowing what the outer rings mean or how to interpret the figure without carefully reading the paragraph before it.

Answer: Thanks for your instructive comments.

In fact your guess is correct, the sizes of the circles represent the service capacity. As mentioned in the previous section, it is assumed that enterprise A is more capable than enterprise B. Thus in the complete competition scenario, the size of the circle of enterprise A is bigger than that of enterprise B. In the co-opetition scenario, as there is transfer of residual capacity, the sizes of circles are the same. In order to make readers understand more clearly, we added sentence to explain to sizes of circles as follows:

“The sizes of circles represent the service capacity of enterprises. The larger the circle, the more capable the enterprise.”

For the meaning of the outer rings, we added sentence to explain as follows:

“The circle with outer ring represents the enterprise's own better profit choice under the current decision of the other enterprise.”

Please see Lines 431-433 on Page 14 for more details.

We hope to have answered Comment #15.
